Behavioral responses of different reproductive statuses and sexes in Hermetia illucens (L) adults to different attractants

Laksanawimol Parichart 1
Singsa Sukdee 1
Thancharoen Anchana koybio@gmail.com 2
1 Faculty of Science, Chandrakasem Rajabhat University , Bangkok , Thailand
2 Department of Entomology, Faculty of Agriculture, Kasetsart University , Bangkok , Thailand
Lazzari Claudio
Electronic publication date: 2023 Jul 25
Publication date: 2023
Volume: 11
Electronic Location ID: e15701
Received 2023 Mar 29; Accepted 2023 Jun 14
Copyright: ©2023 Laksanawimol et al.
Copyright year: 2023
Copyright holder: Laksanawimol et al.
License: This is an open access article distributed under the terms of the Creative Commons Attribution License, which permits unrestricted use, distribution, reproduction and adaptation in any medium and for any purpose provided that it is properly attributed. For attribution, the original author(s), title, publication source (PeerJ) and either DOI or URL of the article must be cited.
License URL: https://creativecommons.org/licenses/by/4.0/

Keywords: Oviposition behavior, Egg laying behavior, Temporal activity, Egg yields, Diptera, Marking technique, Oviposition preferences

Funding: Kasetsart University Research and Development Institute FF(KU) 14.64 This study was financed by the Kasetsart University Research and Development Institute (KURDI) (Grant No. FF(KU) 14.64). The funders had no role in study design, data collection and analysis, decision to publish, or preparation of the manuscript.

==============================
Background

The odor of various fermented organic materials acts as an attractant for oviposition by gravid females of the black soldier fly (BSF) to find larval food sources. Females display oviposition site selection on various organic materials, but little work has been done on the response to substrate attractants under caged conditions similar to those in a BSF farm production system.

Methods

Fifty of each reproductive status (mated and virgin) and sex (males and females) of BSF adults were marked and then exposed to one of five different oviposition attractants in a transparent acrylic chamber: no substrate (control) plus pineapple, mixed vegetables, okara, and fermented fish to represent fruit-, vegetable-, plant protein-, and animal protein-based substrates, respectively. The frequency of the perching activity on the oviposition apparatus and flying behavior under the LED illumination, including the laid egg weight, were recorded.

Results

The sexually-related activities of BSF adults were clearly observed. A majority of the females preferred to perch on the oviposition apparatus and fly around the illuminated area compared to the very low activities of the mated males. The BSF adults displayed different behavioral responses to the different tested attractants. While active flying was common when using plant protein- and animal protein-based substrates, mated females showed the greatest perching preference for plant-based substrates (fruit and vegetables) and this correlated with the laid egg weight.

Discussion

Egg-laying was more likely to happen on the plant-based substrate than on the animal protein-based substrate. However, the strong smell of the animal protein-based substrate could strongly trigger lekking behavior, which is an important part of mating behavior. This knowledge can support egg trapping in nature and also improve the efficiency of egg production in mass-rearing facilities.

Introduction

Currently, interest in several sources of alternative proteins is increasing as a means to support global food production, including insect proteins, which are utilized for both human food and animal feed (Wood & Tavan, 2022). The larvae of the black soldier fly (BSF), Hermetia illucens (L.) (Diptera: Stratiomyidae), have the ability to convert a wide range of organic wastes into high-quality proteins and fats. These can be used to replace or supplement animal feed to lower rearing costs and improve the quality of products. For chickens, replacement of soybean meal by 50 −100% defatted BSF larvae (BSFL) did not affect the layers’ feed intake, laying performance, egg weights, or feed efficiency (Maurer et al., 2016); however, only a 25% supplementation of the hens’ diet was recommended for optimal laying (Bovera et al., 2018). On the other hand, up to 40% supplementation of defatted BSFL can be used in trout diets (Renna et al., 2017), while feeding non-defatted BSFL to chickens could increase their eggshell thickness (Kawasaki et al., 2019).

Due to the high global demand for livestock feed, many small- and industrial-scale BSF farms are distributed worldwide, such as in South Africa, Canada, China, Malaysia, the Netherlands, and USA (Joly & Nikiema, 2019; Siva Raman et al., 2022). Nevertheless, BSF egg production remains both a challenge and the limiting factor in BSFL production systems. The oviposition behavior of BSF is positively related to the egg yield, and so improving our understanding of this should help mass-rearing facilities to increase production.

Adult BSF can mate during the first 1–5 days (peaking at day 2) and lay eggs during days 3–6 (peaking at day 4) after emerging (Tomberlin & Sheppard, 2002). However, the oviposition duration could be extended when the adults were fed some nutritional diets (Macavei et al., 2020; Thinn & Kainoh, 2022). Comparison of different diets revealed that a mixture of pollen, honey, and water could prolong the oviposition duration to 25–46 days (peaking at days 5–7) resulting in twice the number of eggs being produced compared to that with non-fed adults (Thinn & Kainoh, 2022). In addition, several abiotic factors can affect the oviposition response of the insect, such as humidity, temperature, and time of day (Tomberlin & Sheppard, 2002), light sources (Liu et al., 2020; Nakamura et al., 2016; Zhang et al., 2010), light color (Klüber et al., 2020), light duration (Hoc et al., 2019), light intensity (Park et al., 2016), oviposition material (Boaru et al., 2019; Julita et al., 2021) and color (Romano, Fischer & Egnew, 2020), and oviposition substrate (Ewusie et al., 2019; Nyakeri et al., 2016; Nyakeri et al., 2017; Park et al., 2016; Pei & Siong, 2020).

Organic materials play an important role as oviposition attractants for gravid BSF females. A variety of oviposition substrates have been tested to determine the most effective substrate to receive large egg masses. However, the majority of these choice-experiments have assessed the trap efficiency in the field, where the adult density is low; thus, the oviposition site preferences varied due to substrate choices and other environmental conditions. High trap performances were found in vegetable-fruit waste and mashed maize grain (Nyakeri et al., 2016), fruit waste (Sripontan et al., 2017), millet porridge mash (Boafo et al., 2022), and piggery dump waste (Ewusie et al., 2019). A laboratory test of oviposition site selection applied to different maturity stages of bananas revealed that more ripened bananas with a strong attractive odor could be more attractive (Pei & Siong, 2020). Moreover, the presence of BSF pupal cases had the potential to trap BSF eggs. Similarly, Zheng et al. (2013) examined whether the presence of BSF eggs is attractive to oviposit eggs without any substrates or on sterile substrates.

Although information on the oviposition preference of BSF is available, little work has been done on the response to substrate attractants under caged conditions that can be applied to BSF farm production systems. Information about the reproductive performance of BSF is still needed. Thus, this study aimed to evaluate the attractiveness of four oviposition substrates, categorized as fruit-, vegetable-, plant protein-, and animal protein-based substrates, to different sexes (male and female) and mating statuses (virgin and mated) of BSF adults under caged conditions. Additionally, the behaviors of the adults were defined to help understand the reproductive activities.

Materials & Methods

Stock rearing of BSF

Hermitia illucens (BSF) were maintained at the Department of Entomology, Kasetsart University, Bangkok, Thailand, from original colonies purchased from a farmer in the Khon Kaen province in 2019. The BSFL were reared in a blended mix of vegetables (cabbage, romaine, and lettuce) and maintained under natural ambient conditions at approximately 28–34 °C, 70–90% RH, and a 13:11 h L:D cycle. When they became pupae, they were moved to a mesh room (4 (L) × 3 (W) × 3 (H) m), where adults emerged and mating happened. The adults were fed a 30% (w/v) brown sugar solution ad lib. Oviposition devices were composed of a tray of attractant substrate and pieces of wooden sheets above the tray as oviposition materials, and were prepared in the adult cage.

Preparation of BSF adults for experiment

Approximately 1,000–3,000 large (approximately 19–23 mm length) pupae were gently sorted from the stock rearing with a pair of forceps and put into 32-ounce plastic cups with aerated lids. After emergence, the adults were identified for sex by visual inspection as reported (Julita et al., 2020), to prepare virgin males and virgin females. The emerged adults were seldom found to be mating in the small cups due to their required mate-in-flight behavior. Mated adults were obtained from mating pairs collected inside the adult mesh room during the peak time of mating activities, 08.00–13.00 h (Julita et al., 2020). Each mating pair (seen in reverse coupling position) was covered with an 80 mL transparent cylindrical box and collected after separation to avoid disturbance; they then served as mated males and mated females. The ages of the mated males and females were mixed (range 2–4 d) due to random selection from the stock colony. According to Tomberlin & Sheppard (2002), most BSF display mating behavior at day 2 after emergence, and most mating pairs are first-time matings (Permana, Fitri & Julita, 2020); thus, their ages were probably broadly similar.

Insect marking technique

Nontoxic water-based acrylic pens (UNI Posca, PC-5M, and Mitsubishi Pencil (Thailand) Co., Ltd.) were used to mark the thorax of adult flies with different reproductive statuses (mated and virgin) and sexes (males and females) with four contrasting colors that were easily distinguishable from the others: blue, pink, yellow, and silver (Jones & Tomberlin, 2020). Marking equipment was applied from a 50-mL syringe by cutting off the injection end and replacing it with a transparent plastic film with holes as marking areas (Fig. 1A). The seal of the plunger was removed and glued with a piece of sponge cloth (Scotch-Brite™), where a few drops of water were placed to feed the flies. While the flies were feeding, their pronotum positions were in the marking area and were easy to mark. Fifty flies from each category were carefully marked and placed in separate boxes before being released into the experimental cages.

Figure 1 Schematic diagrams.

(A) Marking BSF adults of different reproductive statuses with water-based acrylic paint pens. (B) An experimental cage. (C) An oviposition apparatus composed of wooden sheets as oviposition materials and an oviposition attractant box containing different test attractants.

Experimental design

All behavioral observations were conducted in a transparent acrylic chamber (Fig. 1B) (50 (L) × 50 (W) × 60 (H) cm) with two aeration windows (12 × 12 cm) on the top and bottom and a sliding door opening (21 × 17 cm) on one side to allow loading of flies and equipment. The chamber was settled on a 1-m table with a ventilation fan below. A 1000-watt light-emitting diode (LED) was placed on the top to provide illumination (6500 K, 9000 lm, IP65; Racer Electric (Thailand) Co., Ltd., Samutsakorn, Thailand) and was controlled by a timer for a 13:11 h L: D cycle. The experimental chamber contained an oviposition attractant box with a ventilation lid at the center. Since wooden sheets are the best material for BSF oviposition (Julita et al., 2021; Boaru et al., 2019), five pieces of wooden sheets with small gaps between them were placed above the tray as oviposition materials (Fig. 1C). With respect to the fly diet, four pieces of sponge cloth soaked in 30% (w/v) brown sugar solution were placed at each corner. Due to the clearly observed fly behavior, the lek position of the males was not prepared inside the chamber. The ambient temperature in the chamber ranged from 26–37 °C. In total, 250 marked BSF adults of different reproductive statuses and sexes were released into the experimental chambers to examine the no-choice oviposition tests of BSF on each substrate in the two available chambers. Six replicates were used for each treatment.

Four different oviposition attractants: pineapple, mixed vegetables, okara, and fermented fish (as fruit-, vegetable-, plant protein-, and animal protein-based substrates, respectively) plus a no substrate control were tested in the experiment. Prior to use, 300 g of each substrate was fermented for 2 d to increase their attractive smell and then mixed with 50 mL of fermented effective microorganisms (EM Extra Co., Ltd., Bangkok, Thailand), except for the fermented fish, which was prepared at 350 g without fermentation. The fermented effective microorganisms help inhibit the growth of some pathogenic bacteria (Lee, Yun & Goo, 2022).

Observations of BSF adult behaviors were categorized into two activities: (1) perching on the oviposition apparatus (an attractant box and wooden sheets) and (2) flying under the illumination. We were unable to identify individuals from the different markings (just the male/female and virgin/mated statuses) and so the frequency of perching and flying events in each reproductive status and sex were recorded. The behaviors were recorded for 5 min every 15 min from 10.00 h to 16.00 h for a 2-day period that covered the peak of the laying egg behavior (at four days old). In total, 25 observational data points were received per day. After behavioral observation, the experimental chamber was continuously monitored for egg masses at days 3 and 5 to avoid hatching on the oviposition apparatus. The laid eggs were gently removed from wooden sheets using a cutter and then weighed on an analytical balance (OHAUS Pioneer PA214 Analytical Balance, USA).

Data analysis

The temporal activities of the BSF adults were grouped into five time periods. Differences in temporal activities, behaviors, and reproductive statuses were analyzed via one-way analysis of variance (ANOVA), followed by Fisher’s LSD test for multiple comparison tests. Pearson correlation was used to determine associations between the number of perching adults and egg weights. For hypothesis testing, a critical value of α = 0.05 was used. The data are displayed as the average ± one standard error (SE). All statistical analyses were performed via SPSS, version 14 (SPSS for Windows, Chicago, IL, USA).

Results

Temporal activity

A total of 1,500 observational data points of BSF activities were observed from 10.00 h to 16.00 h. The BSF in each chamber exhibited perching behavior on the walls and floor of the chamber, as well as on the oviposition apparatus, and a flying behavior. These spiral flight activities took place under the lighting. Temporal variability in the BSF behaviors appeared during the daytime, with the majority of BSF adults displaying a flight behavior more than perching on the oviposition apparatus. However, the peak of both adult activities occurred in the morning and decreased obviously between 15.00 h and 16.00 h (F = 2.536, df = 4, P = 0.041 for the flight behavior and F = 4.180, df = 4, P = 0.003 for the perching behavior) (Fig. 2).

Figure 2 The temporal activity (times/5 min) of BSF adults displaying different behaviors.

Data are shown as the mean ± 1SE, derived from six trials. Letters indicate significant differences among time periods (P < 0.05).

Sexual-related activity

In order to understand the sexual behavior of adult BSF, we investigated the behavioral responses of the two reproductive statuses and two sexes of BSF to oviposition attractants. Based on the observations over 2 d, the perching activity was significantly lower on day two than on day 1 (t = 3.064, df = 248, P = 0.001), whereas the flying activity was significantly higher at day 2 (t =  − 2.760, df = 184.145, P = 0.006). Sexually-related activity was observed (Fig. 3), and was significantly higher in females (more than 60%) than in males. Both mated and virgin females preferred to perch on the oviposition apparatus and fly around the illuminated area compared to the very low activities of mated males. Only 15–20% of BSF activities were performed by males. Surprisingly, the majority of flying adults were virgin females. The number of virgin females that displayed both perching and flying behavior correlated with the number of males (r = 0.093, P = 0.001 for the virgin perching male; r = 0.258, P = 0.000 for the mated flying males).

Figure 3 Sexually-related activities of BSF adults with different behavioral responses observed (times/5 min) in day 1 and day 2 of the experiments.

(A) Perch on oviposition apparatus; and (B) fly under the light. Data are shown as the mean ± 1SE, derived from six trials. Letters indicate significant differences among reproductive statuses (P < 0.05).

Behavioral responses to oviposition attractants

The BSF adults displayed significantly different behavioral responses in the presence of the five different tested attractants (Fig. 4A). The plant protein- (okara) and animal protein- (fermented fish) based substrates with stronger smells attracted flying adults more than the other substrates (F = 6.043, df = 4, P = 0.000). The control group (no substrate treatment) showed the least amount of flying behavior. For the perching behavior, the different substrates displayed almost the same results. The substrate with the least attractiveness was the mixed vegetables, which displayed the same results as the control (F = 5.361, df = 4, P = 0.000). Based on both activities, okara, fermented fish, and pineapple were the top three attractive substrates to BSF adults.

Figure 4 Behavioral responses (times/5 min) of BSF adults to various oviposition attractants.

(A) All reproductive statuses; (B) flight activity of each reproductive status; and (C) perching on the oviposition apparatus of each reproductive status with averaged egg weight found on each oviposition attractant (line graph). Data are shown as the mean ± 1SE, derived from six trials. Letters indicate significant differences among reproductive statuses (P < 0.05). Letters in shade indicate a significant difference in egg weight among oviposition attractants (P < 0.05).

From Figs. 4B and 4C, observing the BSF in each reproductive status makes it clear how the flies responded to the attractants. In all substrates, the frequency of flying adults was ranked (highest to lowest) as virgin females >mated females >virgin males >mated males; however, many more virgin males showed the flight behavior in response to the mixed vegetables substrate than to the other substrates. Similarly, the presence of perching BSF on the oviposition apparatus displayed the same sexual activities with a high number of both virgin and mated females. The mated females preferred to stay near the attractants, in contrast to the control group, which had the lowest number. Similarly, high numbers of virgin females were found on the oviposition apparatus in the control group, even though there was no attractant substrate.

Egg yields

In order to determine the performance of each attractant, egg masses were collected twice on days 3 and 5 of each experiment (Fig. 4C). All the eggs were found in the gaps between the wooden sheets in all experiments. The lowest amount of egg masses were found in the control group, which had the lowest oviposition attraction performance. The highest average weight of egg masses were found in the fruit-(pineapple) and vegetable-based substrates (0.718 ± 0.148 g and 0.642 ± 0.068 g, respectively). The frequency of mated females on the oviposition apparatus correlated with the laid egg weight (r = 0.494, P = 0.012), but not the number of overall perching adults (r = 0.073, P = 0.746).

Discussion

Light intensity is an important factor that affects the mating and oviposition behavior of BSF adults. Since the light intensity varies according to the time of day, mating behaviors were also expected to vary temporally. The peak mating behavior was reported to be between 11.00 and 12.00 h, when the light intensity was at its maximum, while the peak oviposition happened later, at 13.00–14.00 h (Julita et al., 2020). On the other hand, mating peaked in Wuhan, China, at 10.00 h, while the maximum light intensity was at 13.00 h but the flies displayed different behavior when exposed to a quartz-iodine lamp, which has same spectrum as sunlight, when mating peaked at 11.00 h (Zhang et al., 2010).

In this study, artificial lighting (LED) was used in the caged experiment to evaluate the BSF oviposition preferences. Although artificial illumination is not as effective as natural lighting (Nakamura et al., 2016; Zhang et al., 2010), it can be controlled under caged conditions. Under LED lighting, the appearance of mating pairs in the experimental cage was reported to rarely be due to a lower light intensity, but rather higher light intensities encouraged the mating activity (Tomberlin & Sheppard, 2002). Park et al. (2016) suggested that the oviposition rate also depended on the light intensity; BSF adults preferred to lay eggs in sunny conditions more than in the shade. However, artificial lighting was found to have no effect on the number of clutches or eggs per female (Nakamura et al., 2016). In the absence of light at a wavelength from 332 to 535 nm, unsuccessful mating and laying of infertile eggs can still take place (Heussler et al., 2018). Nevertheless, the fertility of eggs was not examined in the present study.

In this study, 50 mated females were released into the cage, and the oviposition behavior happened normally when compared with the number of eggs in a previous work (Julita et al., 2020). Although the mating and oviposition behaviors of BSF have been recognized by researchers for several decades, there is little information about their flight behavior, especially that of females, whereas the effects of artificial lighting on the behavior of several other dipteran species is available. In the case of tephritids, light was shown to play a significant role in the formation and location of leks (Arita & Kaneshiro, 1989). It is possible that a similar trend exists in BSF because BSF males in the laboratory cage preferred to select lek sites with a high amount of light.

On the second day after emerging, BSF adults exhibited courtship behavior, and the males performed lek behavior by congregating on leaves (called the lekking area), flying together, and defending territories to compete for a successful female choice (Julita et al., 2020; Tomberlin & Sheppard, 2002). At the beginning of the courtship behavior, 80% of females displayed flight behavior to attract a lekking male, who then approached her during flight and grasped her thorax in a dorsal mounting position before landing and copulating afterwards (Giunti et al., 2018). Similarly, this study’s findings revealed that females displayed a flying behavior twice as frequently as males did, especially for virgin females. Although female BSF have been reported to be monogamous (Giunti et al., 2018; Tomberlin & Sheppard, 2002), they have also been reported to remate within 4–5 days after the previous mating, gaining a higher genetic diversity in their progeny but with no change in the number of eggs, egg weight, or egg fertility (Permana, Fitri & Julita, 2020). Thus, the results of this study may include many mated as well as virgin females flying in the experimental cage. In the sand fly (Lutzomyia longipalpis), a high proportion of virgin females were attracted to male pheromone and host odor in the same location to receive the opportunity to mate and blood-feed, whereas females did not attract the males (Bray & Hamilton, 2007). Several males performed a standing behavior, which is difficult to distinguish between lekking and just remaining motionless; however, some of them participated in the female flight group, and so probably wanted to have multiple matings. However, as the males are normally active under light, they probably displayed less flight in the experimental set-up because of the limited height of the experimental cage.

A gravid female can lay 450 −600 eggs in a 6.5–16.2 min period (Julita et al., 2020). The requirements for BSF oviposition are a dry oviposition site with a crevice or a small gap and an odor from decaying organic substances. During egg-laying, females display oviposition site selection on different organic materials (Boaru et al., 2019; Julita et al., 2021) and attractant substrates (Ewusie et al., 2019; Nyakeri et al., 2016; Nyakeri et al., 2017; Park et al., 2016; Pei & Siong, 2020), which are beneficial for avoiding predators and serving as a source of nourishment for the developing larvae after hatching (Julita et al., 2020). However, the substrates most preferred by the gravid females are not the best for larval development (Boafo et al., 2022), although the nutrient composition of the substrate can relate to the larval growth (Barragan-Fonseca, Dicke & Van Loon, 2018; Cammack & Tomberlin, 2017; Gold et al., 2020).

Different compositions of substrates are likely to affect the BSF attraction and oviposition preferences. The different attractiveness of substrates to gravid females was found to be influenced by various volatile organic compounds (VOCs) produced by bacteria and fungi (Scieuzo et al., 2021). For instance, Staphylococcus sp. produced 2-methyl-butanal, which was used as a specific oviposition cue by gravid females and then the 2-methyl-butanal concentration decreased with increasing larval activities. According to the findings of a recent study, not only gravid or mated females locate to the oviposition sites, but the majority of virgin females and some males are also attracted to decaying organic substances. It is possible that lekking male groups and virgin females will also be drawn to VOC-releasing areas, allowing for courtship and oviposition.

A recent study found that mated females showed the greatest preference for pineapple (fruit-based substrate) among the various evaluated substrates, as indicated by both the attractive response on the oviposition apparatus and the number of deposited eggs. This result corresponded with previous studies (Table 1) that found that plant-based substrates (fruits, vegetables, and grains) were more effective than animal protein-based substrates and manure (Boaru et al., 2019; Nyakeri et al., 2016; Sripontan et al., 2017). This difference is most likely due to differences in the composition of the diverse bacterial communities in different substrates, resulting in the production of different compositions of VOCs to attract BSF (Neher et al., 2013). Aside from attracting oviposition, VOCs can be used to increase the attractiveness of interspecific competition for resources with house flies because the levels of VOCs decrease with larval development, resulting in a less attractive substrate for house flies (Adjavon et al., 2021). The symbiotic bacteria that produce VOCs were also found on the laid eggs that can induce oviposition in gravid BSF females (Zheng et al., 2013). Thus, oviposition is influenced by not just the attraction of the substrate, but also the attractiveness of the eggs themselves. As a consequence, adding larval frass into the oviposition substrate could enhance female BSF oviposition (Nyakeri et al., 2017) and deter oviposition by house flies, the main larval competitor species (Adjavon et al., 2021).

Table 1 Evaluation of different oviposition attractant substrates, which were categorized into six groups; fruit-based substrates (FS), vegetable-based substrates (VS), plant protein-based substrates (PPS), animal protein-based substrates (APS), manure (M), and food waste (FW).

Those selected for evaluation are shown in the circular areas. The shaded symbol represents the most effective substrates in the experiments.

Tested oviposition attractant substrates	Effective substrates	References	
FS	VS	PPS	APS	M	FW			
●	●	●	○	○	○	Mashed maize grains	Nyakeri et al. (2016)	
						Vegetable wastes		
○			○	●	●	Cow manure	Nyakeri et al. (2017)	
●				○	○	Mixture of fruits; wax apple, pineapple, apple, water melon, and melon	Sripontan et al. (2017)	
			○		○	20% Calf feed mixture	Park et al. (2016)	
○						Over-ripened banana	Pei & Siong (2020)	
○	○	●		○		Millet porridge mash	Boafo et al. (2022)	
				○		Piggery waste dump	Ewusie et al. (2019)	

Although strong-odor oviposition substrates, such as carcasses, fresh meat, and internal organs of animals, are widely believed to be the best (most effective) attractants for BSF oviposition, they are also attractive to other fly species, such as Calliphoridae, Sarcophagidae, and Muscidae (D’Almeida & Fraga, 2007; Sukhapanth, Upatham & Ketavan, 1988). Moreover, these other fly species colonize carcasses earlier than BSF and so could also be larval competitors (Cruz et al., 2021). Thus, the use of fresh animal protein-based substrates in field trapping is limited due to intraspecific competitors and the production of human disease vectors. On the other hand, the sugar in fermented fruit and vegetables could attract another phorid fly, Megaselia scalaris (Reguzzi et al., 2021), which is also an important competitor and has been found at 38% (by number) in a fermented coconut waste substrate at pH 4–5 (Hasan & Dina, 2019). Thus, the addition of conspecific bacteria from BSF eggs or larval frass, as mentioned above, might be a good solution to improve the efficiency of oviposition substrates.

Our data demonstrated that certain fruit- and vegetable-based substrates could be the most attractive for gravid females to lay eggs. On the other hand, plant protein- and animal protein- based substrates (okara and fermented fish, respectively) are interesting substrates. The strong-odor of okara and fermented fish is not particularly appealing for oviposition, but it can strongly induce a lekking behavior, which is an important mechanism of mating behavior. Thus, from our findings we suggest that the selection of optimal substrates for both behavior responses—lekking and oviposition behavior—would improve the efficiency of egg production, especially egg trapping in nature.

Conclusions

Insects, particularly dipterans, used specific chemical cues to increase their fitness. Bacterial communities in the decomposing (waste) substrate produce specific VOC profiles that are associated with the insects’ behavioral responses. This study provides preliminary information on the activities of BSF adults of different reproductive statuses and sexes in response to different substrates, in order to apply this knowledge to improve the efficiency of egg-trapping in commercial rearing of BSF. Sexual-related activities were observed in BSF during courtship, mating, and oviposition. During the lekking behavior of males, most females (both virgin and mated females) displayed flight behavior to choose a male to mate with, while some males joined in the flight aggregation. Most mated females were found on the oviposition substrate to lay eggs. The weight of laid eggs correlated with the frequency of perched mated females.

The plant protein- and animal protein-based substrates (okara and fermented fish, respectively) were found to be more attractive substrates to induce flying females, which might be related to the lekking behavior of the males. The female BSF perching on the oviposition apparatus were mostly mated females. Plant-based substrates were highly attractive, while pineapple was the best oviposition attractant.

Supplemental Information

File S1 Behavioral data

Click here for additional data file.

The authors thank the Royal Project Foundation for giving us the vegetable wastes that were used in the experiment to raise the BSFL stock. We also thank Miss Kanyanat Khaekratoke and Miss Narathip Taweelas for helping us in the lab.

Additional Information and Declarations

Competing Interests

Author Contributions

Data Availability

The authors declare there are no competing interests.

Parichart Laksanawimol conceived and designed the experiments, performed the experiments, analyzed the data, authored or reviewed drafts of the article, and approved the final draft.

Sukdee Singsa performed the experiments, analyzed the data, prepared figures and/or tables, and approved the final draft.

Anchana Thancharoen conceived and designed the experiments, performed the experiments, analyzed the data, prepared figures and/or tables, authored or reviewed drafts of the article, and approved the final draft.

The following information was supplied regarding data availability:

The raw measurements are available as a Supplemental File.

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
