# Peer review of "Behavioral responses of different reproductive statuses and sexes in Hermetia illucens (L) adults to different attractants"

_PeerJ, doi:10.7717/peerj.15701_

## Round 0.1 · original submission · Major Revisions

Both reviewers are very appreciative of your contribution but also pointed out a number of points that need your attention.

·

Basic reporting

Paper submitted by Laksanawimol et al present original data on the preference of black soldier flies for different substrates. Authors tested experimentally the activity and oviposition of adults of different status and sexes.
The paper is understable, it cites various and valuable references, although I think that three papers from an unique team are more cited, namely Julita et al 2020, 2021 and Permana et al 2020, which are not easily accessible out from Asia.
Figures and tables are clear
Conclusions are supported by results, despite some diffrences in tested substrates which are not as clear as the authors claim.

Experimental design

This is not the first test of prefered substrate for BSF oviposition, and certainly not the last, but it adds some pieces in the puzzle of BSF farming.
Experimental design appears rigorous but lacks some clarity, especially for the status of adult flies.
One question is that authors ask the need for experiments in farm conditions (line 86) but conducted their experiments in lab conditions, which are very different.
Techniques are accurate, even if the labelling of flies by colour seems to me very complicated; Usually, flies can be handled with fingers and labelled with a posca or a droplet of acrylic colour without any apparatus.
Lines 108-109, not clear how flies are sampled, by forceps, that is for me one by one, and it is writen 1000 to 3000, that is not one by one?
Lines 154-155, what are "effective microorganisms", is it a special mix for BSF? If yes, does it really works?
Concerning the status of adults, I suggest to sort the text by status and sex, and to clearly separate traits that were recorded, ie Activity, flight, perching, mating, oviposition...

Validity of the findings

Results are presented before being discussed. All data a presented and staistically tested,
Line 207, how the comparison of two samples gave a Df=4?
Some points are not discussed:
Is a male activity related to female activity instead of substrate?
Are egg masses related to the number of egg laying females or to the egg load of individual females? It is suggested by the relationship between the number mated females on a substrate and the weight of eggs collected. Moreover, could egg mass weight being a way to know the number of egg laying females?
Were egg fertile?
Discussion:
I think that considerations of the lekking system are out of the purpose, because the experimental design do not allow to identify male territories, and the timing of observations - not continuous - allows to check for copulations which are long lasting but not for shorter behaviours as are courtships.
Line 292, VOCs were not tested here, they can be adressed but not for a so long paragraph.

Additional comments

The over citation of Permana and Julita works does not allow a generalization of the results to various farming or experimental situations.

Reviewer 2 ·

Basic reporting

This is an interesting study, especially, the observation of the ovipositing behavior of BSF. The information is very useful for mass rearing. The paper is generally well written and structured. However, in my opinion the paper has some points to clarify.
Below I have provided some suggestions.
1. Line 147: “In total 200 marked BSF adults…..” the number of each treatment should be added (50 insects for each?).
2. Line 161-162: “the behaviour were recorded …..for 2 days peroids” why the author observed for 2 days? Is it the critical period for oviposition?
3. Why virgin females are high activity than other reproductive status? Compare to other insect, This behavior similar to other insects?
4. Figure 2-4: Please indicate unit for the frequency in Y axis or in the figure caption (times/5 min?).
5. Figure 3: Please check the symbol on virgin female (day 1=b , day2= a or day 1=a , day2= b?)
6. According to Figure 3B, why “frequency of fly under the light” on day2 is significantly decrease for male.
7. Figure 3: please indicate Figure 3A and B in the figure caption.
8. Figure 4: the details of “trend line” should be added in figure caption

Experimental design

no comment

Validity of the findings

no comment

Additional comments

no comment

---

## Round 0.2 · Minor Revisions

Thank you very much for the new improved version of your manuscript, acknowledging the comments of the reviewers.

Rev#1 requested "...to sort the text by status and sex..", and this was probably the reason you included "...and sex." in the title and the text. The result is that the title and some passages turned into strange statements when related to "oviposition preferences" because only females lay eggs. I recommend reviewing these passages, in order to clearly distinguish "oviposition preferences" from "oviposition-related cues".

---

## Round 0.3 · accepted · Accept

Thank you for improving the clarity of the manuscript.

·

Basic reporting

Paper was corrected as recommanded

Experimental design

ok

Validity of the findings

ok

Additional comments

ok